# Accidental Hypothermia: 2021 Update

**DOI:** 10.3390/ijerph19010501

**Published:** 2022-01-03

**Authors:** Peter Paal, Mathieu Pasquier, Tomasz Darocha, Raimund Lechner, Sylweriusz Kosinski, Bernd Wallner, Ken Zafren, Hermann Brugger

**Affiliations:** 1Department of Anesthesiology and Intensive Care Medicine, St. John of God Hospital, Paracelsus Medical University, 5020 Salzburg, Austria; 2International Commission for Mountain Emergency Medicine (ICAR MedCom), 8302 Kloten, Switzerland; Mathieu.Pasquier@chuv.ch (M.P.); kenzafren@gmail.com (K.Z.); hermann.brugger@eurac.edu (H.B.); 3Department of Emergency Medicine, Lausanne University Hospital, 1011 Lausanne, Switzerland; 4Department of Anesthesiology and Intensive Care, Medical University of Silesia, 40-001 Katowice, Poland; tomekdarocha@wp.pl; 5Department of Anesthesiology, Intensive Care Medicine, Emergency Medicine and Pain Therapy, Military Hospital, 89081 Ulm, Germany; raimund.lechner@uni-ulm.de; 6Faculty of Health Sciences, Jagiellonian University Medical College, 34-500 Krakow, Poland; sylweriusz.kosinski@uj.edu.pl; 7Department of Anesthesiology and Critical Care Medicine, Medical University of Innsbruck, 6020 Innsbruck, Austria; bernd.wallner@tirol-kliniken.at; 8Department of Emergency Medicine, Alaska Native Medical Center, Anchorage, AK 99508, USA; 9Department of Emergency Medicine, Stanford University Medical Center, Stanford University, Palo Alto, CA 94304, USA; 10Institute of Mountain Emergency Medicine, Eurac Research, 39100 Bolzano, Italy; 11Department of Anesthesiology and Intensive Care Medicine, Medical University of Innsbruck, 6020 Innsbruck, Austria

**Keywords:** accidental hypothermia, cardiac arrest, cardiopulmonary resuscitation, emergency medicine, extracorporeal life support, rewarming

## Abstract

Accidental hypothermia is an unintentional drop of core temperature below 35 °C. Annually, thousands die of primary hypothermia and an unknown number die of secondary hypothermia worldwide. Hypothermia can be expected in emergency patients in the prehospital phase. Injured and intoxicated patients cool quickly even in subtropical regions. Preventive measures are important to avoid hypothermia or cooling in ill or injured patients. Diagnosis and assessment of the risk of cardiac arrest are based on clinical signs and core temperature measurement when available. Hypothermic patients with risk factors for imminent cardiac arrest (temperature < 30 °C in young and healthy patients and <32 °C in elderly persons, or patients with multiple comorbidities), ventricular dysrhythmias, or systolic blood pressure < 90 mmHg) and hypothermic patients who are already in cardiac arrest, should be transferred directly to an extracorporeal life support (ECLS) centre. If a hypothermic patient arrests, continuous cardiopulmonary resuscitation (CPR) should be performed. In hypothermic patients, the chances of survival and good neurological outcome are higher than for normothermic patients for witnessed, unwitnessed and asystolic cardiac arrest. Mechanical CPR devices should be used for prolonged rescue, if available. In severely hypothermic patients in cardiac arrest, if continuous or mechanical CPR is not possible, intermittent CPR should be used. Rewarming can be accomplished by passive and active techniques. Most often, passive and active external techniques are used. Only in patients with refractory hypothermia or cardiac arrest are internal rewarming techniques required. ECLS rewarming should be performed with extracorporeal membrane oxygenation (ECMO). A post-resuscitation care bundle should complement treatment.

## 1. Introduction

Accidental hypothermia is the involuntary drop of the core temperature below 35 °C [1]. Accidental hypothermia may be caused in a healthy subject by exposure to a cold environment (primary accidental hypothermia) or triggered by other conditions, most commonly illness, intoxication, or trauma, called secondary hypothermia (Table 1). The pathophysiology of accidental hypothermia in injured or ill patients has been described in detail elsewhere [2]. This article does not cover therapeutic hypothermia, which is intentionally-induced hypothermia used in fields such as cardiac surgery, and as part of a post-resuscitation care bundle after cardiac arrest in neonates and adults.

As accidental hypothermia develops, vital signs decrease until cardiac arrest occurs. Hypothermia has detrimental effects on the function of multiple organs, including heart, brain, kidney, blood coagulation, and, possibly, the immune system [4,5]. Overall, accidental hypothermia increases morbidity and mortality in affected victims. The aim of this work is to provide an up-to-date overview on clinical aspects of accidental hypothermia.

## 2. Epidemiology

Accidental hypothermia has been recognised since ancient times [6]. Throughout history, hypothermia has been a disease of war and disasters such as avalanches, earthquakes and tsunamis [7,8,9]. Nowadays in developed countries, primary hypothermia affects mainly people who live, work, and recreate outdoors putting themselves at risk in cold environments and homeless people. In less developed countries primary hypothermia affects homeless people and people in mass accidents; for example, victims of avalanches burying villages and travellers in poorly protected mountain areas. In general, the risk of accidental hypothermia from exposure to cold increases with decreasing temperature, but many cases among homeless individuals occur during periods of low and moderate cold stress [10]. Secondary hypothermia has been recognised as a phenomenon in elderly and patients with multiple comorbidities, mainly in Japan, which has the oldest population worldwide [11]. Countries with comparably old populations may see similar rises in coming decades. In the United States, primary hypothermia is the cause of at least 1500 deaths a year. From 1995 to 2004 about 15,000 patients presented annually to hospitals with hypothermia and other cold-related conditions [12]. The incidence of accidental hypothermia in European countries and New Zealand ranges from 0.13 to 6.9 cases per 100,000 per year [13,14,15,16]. Hypothermia is responsible for about 2 deaths per 100,000 per year in Scotland and 5 per 100,000 per year in Poland [17,18]. Frequencies vary widely because for most countries no reliable national data are available. No epidemiological data are available from less developed countries in Africa, South America, and Southeast Asia.

## 3. Pathophysiology

Humans are homeothermic. Core temperature is tightly regulated with minimal diurnal variations at about 37 ± 0.5 °C. Central (hypothalamic) and peripheral thermoregulation (peripheral vasoconstriction and dilation, shivering, and sweating) regulate the core temperature autonomically (Figure 1). Humans may consciously influence core temperature by behaviour, exercise, and clothing.

In a healthy individual, hypothermia can be the result of excessive transfer of energy to a cold environment through conduction, convection, evaporation, or radiation (primary hypothermia). Hypothermia can also be caused by conditions that impair thermogenesis or thermoregulation (secondary hypothermia, Table 1). Children, and small adults with low body mass indices (BMIs) are most susceptible to hypothermia because of their large body surface to weight ratios, allowing greater heat loss compared to larger individuals with normal or high BMIs [22]. Mild or moderate hypothermia can occur in in urban and rural areas in cold and moderate climates during all seasons [11,23,24]. Natural disasters can place large numbers of victims at risk for accidental hypothermia [25]. Severe hypothermia is common in temperate or cold climates as well as in mountain, areas. Predisposing factors include cold and wet environments, fatigue, exhaustion and high altitude, with hypoxia. In one study of mountain areas, 57% of severely injured casualties (injury severity score—ISS ≥ 16) and more than a third of severely injured casualties with traumatic brain injury were hypothermic at hospital admission [26]. In another study, over 90% of trauma patients trapped in motor vehicles were hypothermic [27]. Especially in trauma patients, hypothermia can cause a substantial increase in mortality and complications from decreased cardiac contractility, dysrhythmias, trauma-induced coagulopathy with increased risk of bleeding, and diminished inflammatory response [28,29]. A retrospective European trauma registry study found that 56% of patients at least 16 years, old for whom temperature data were available, with an injury severity score (ISS) ≥ 9 and a core temperature below 33 °C, seen in the emergency department who died or were admitted to an intensive care unit had multiple organ system failure. The overall mortality of this group was 32% [30]. Patients who were transferred directly to the ICU and were not seen in the emergency department and patients with isolated head injuries were excluded. Since hypothermia is more often found in severely injured patients, it is difficult to determine if there is a causal relation between hypothermia and increased mortality [28]. 

In Japan, hypothermia is a common problem indoors for elderly people with multiple comorbidities living alone. especially as complication of injury, illness or use of medications [31]. In many countries, hypothermia affects and often kills the most disadvantaged members of society, such as homeless people and those addicted to alcohol or drugs. 

Hypothermia causes a decrease in vital signs and can lead to cardiac arrest (Table 2). A windy and wet environment will speed cooling. The wind chill index describes the combined effect of ambient air temperature and wind speed on the skin surface temperature [32]. In avalanches, the cooling rate of buried victims may reach 9 °C/h [9]. The cooling rate of hyperthermic patients during immersion in water at 1–2 °C can reach 5 °C/10 min [33]. The cooling rate of normothermic or hypothermic patients in cold water depends heavily on conditions. It is much slower, but still significant.

A person immersed in cold water (<15 °C) can arrest after 30 min [35]. After extrication from a cold environment, cooling continues by redistribution of the heat from the warm core to the cold periphery primarily by countercurrent heat exchange in the peripheral circulation but also by conduction of heat from the core to cooler surface tissues. Shivering may be decreased or abolished in patients who are exhausted, sedated, or critically ill. Without shivering, core temperature drops more rapidly, and spontaneous rewarming is likely to be impossible. Additional factors that may contribute to rapid cooling are sweating, impaired consciousness, wearing only a single thin layer of clothing, failure to cover the head, and a slender body habitus with limited subcutaneous fat. During avalanche burial, the absence of an air pocket and development of hypercapnia can also lead to rapid cooling [9].

In young, healthy adults, hypothermia-induced cardiac arrest may occur below 30 °C. In elderly patients and those with comorbidities, the myocardium may become more irritable and hypothermia-induced cardiac arrest may be triggered below 32 °C [34]. Vital signs may still be present in patients with core temperatures below 24 °C [36]. Hypothermic cardiac arrest is fundamentally different than normothermic cardiac arrest. Treatment and outcomes differ substantially [37]. 

Cardiocirculatory collapse during extrication or transfer of hypothermic patients is referred to as rescue collapse. This can manifest as a witnessed cardiac arrest [38]. The underlying mechanism is not well understood and may be multifactorial. In hypothermic patients, rescue collapse may be coincidental during rescue, but is more likely to be caused by hypovolemia, cardiac dysrhythmias triggered by interventions, such as central venous catheterisation, biochemical changes, or, most frequently, by mechanical stimuli such as sudden movement [38,39]. In hypothermic patients, cardiac arrest may also be caused by afterdrop (further cooling, even after rewarming has started). The main cause of afterdrop is reperfusion of cold body parts during rewarming, but conductive heat transfer between colder and warmer body regions may also play a role. In one study, the mean core temperature of patients with witnessed hypothermic cardiac arrest was 23.9 ± 2.7 °C. Only 2.4% had a core temperature higher than 28 °C. In patients with core temperatures above 30 °C there was no cardiac arrest solely attributable due to hypothermia [38,40]. In general, a normal level of consciousness correlates with a low risk of hypothermic cardiac arrest [41]. Overall, rescue collapse appears to double the risk of death in severely hypothermic patients [40]. Interventions, such as sudden movements of the patient or making the patient exercise, that are known to cause rescue collapse should be avoided. 

## 4. Diagnosis of Hypothermia 

Out-of-hospital diagnosis of accidental hypothermia may be challenging. Most importantly, accidental hypothermia should be considered and ruled out in patients with a history of cold exposure or predisposing conditions (Table 1), and if the trunk feels cold to touch [42]. If core temperature measurement is not available, clinical diagnosis can be made by evaluating the vital signs (Table 2). Vital signs generally decrease linearly as core temperature decreases [43]. Shivering is not a consistent symptom. It can be suppressed for various reasons and should not be used to diagnose the degree of hypothermia. However, if a patient is shivering, core temperature is ≥30 °C. Using the level of consciousness is the best way to assess hypothermia if a core temperature cannot be measured. However, level of consciousness may be influenced by factors other than core temperature, such as intoxication and trauma [3]. The Revised Swiss System should be used if core temperature cannot be measured (Table 3) [41]. 

Definitive diagnosis and determination of the level of hypothermia can only be made by the measurement of core temperature. Because external parts of the body cool more rapidly than the core, core temperature should be measured as close as possible to the vital organs (the brain and heart) [42]. 

The ideal thermometer would be minimally invasive, easy to use, hygienic, independent of environmental conditions, measures core temperature with high accuracy, and has a short response time [42]. No temperature measurement device exists that satisfies all these requirements. All thermometers have spatial and temporal variations in measurement [45]. Only a few studies have assessed core temperature measurement in severe hypothermia in humans and even fewer have assessed core temperature measurement in cold environments [45,46,47,48,49,50,51,52,53,54,55]. 

Table 4 shows the methods of core temperature measurement with their characteristics, and limitations. Peripheral measurements such as skin, so-called temporal artery, oral, and axillary temperatures and infrared tympanic temperatures, are influenced by environmental conditions and are not accurate in hypothermia [42,54,56]. An exception is the thermistor-based epitympanic temperature measurement, using a device intended for outdoor use with good insulation, if the external auditory canal is patent [49,55].

Nonvascular central thermometers, such as oesophageal, urinary bladder, rectal, and nasopharyngeal thermometers, correlate well with pulmonary artery catheter measurement, the most accurate method of measuring core temperature, in the steady state [54,55,73]. During rapid temperature changes, nonvascular central thermometers readings lag behind true core temperature because they are influenced by the surrounding tissue and body contents, such as air, urine, and stool, that have thermal inertia [45,48,54,63]. Nonvascular thermometers are most reliable during a thermal steady state, but readings are still often discordant [54]. If accurate knowledge of true brain temperature is critical, oesophageal or epitympanic temperature should be measured [54]. 

New methods, such as microwave and zero-heat-flux thermometers are being developed for non-invasive temperature measurement. However, they are still experimental and not yet suitable core temperature measurement in deep hypothermia (<28 °C) or in a cold environment [46,47,65]. In practice, a rough initial out-of-hospital estimate of core temperature can be made by touching the patient’s chest and by using the revised Swiss system. If core temperature measurement is available, it is best made using a thermistor-based epitympanic probe in non-intubated patients and using an esophageal probe in intubated patients [42,73]. Out-of-hospital deep nasopharyngeal measurement (10–14 cm) may be an alternative in patients with impaired consciousness and without a secure airway if no thermistor-based epitympanic device is available [54,72]. In hospital, urinary bladder temperature is a widely used alternate method [42]. Infrared epitympanic devices are suitable for screening for hypothermia, but not for monitoring core temperature.

## 5. Treatment

### 5.1. Out of Hospital

Core temperature can decrease during medical care, with exposure, analgesia, anaesthesia, and infusion of fluids [75]. It is critical to prevent or slow further heat loss, and to start rewarming as soon as possible. 

Prevention of further heat loss

Normothermia should be maintained, but prehospital rewarming is not feasible in most emergency medical systems because of limited equipment and short transport times (<1 h). The emphasis should be on preventing further heat loss. Essential measures for hypothermic patients in the pre-hospital setting are extrication from the cold environment, limitation of further heat loss and rapid transfer to hospital [34,39,76]. Patients with impaired thermoregulatory mechanisms and decreased ability to shiver because of conditions such as trauma, secondary hypothermia, use of sedative drugs, or neuromuscular blockade, are at high risk of developing severe hypothermia and hypothermic cardiac arrest [34,39]. Prevention of hypothermia is essential to avoid complications of hypothermia and increased mortality. The principles of pre-hospital management of hypothermia depend on the clinical stage (Table 2 and Table 3). 

Mild hypothermia (Stage 1)

Uninjured patients with mild hypothermia are usually fully alert. They can usually be managed on site without transport to hospital [3,76]. Passive rewarming by removing the patient from the cold environment, optimising insulation, offering food and warm drinks, and promoting active movements, is usually sufficient [3].

Moderate or severe hypothermia (Stage 2 or 3)

Patients with moderate or severe hypothermia require active rewarming [76,77]. The whole body should be insulated to reduce the risk of further cooling [1,3] (Figure 2). 

Protection should be provided against cold, wind, and moisture. A hypothermic patient should be packaged with several layers. A patient who is unable to walk, should be placed in a horizontal position on a stretcher. If dry clothing or layers are available, wet clothing should be cut off in a protected environment. Large chemical or electric heat packs should be applied to the trunk on the chest and back, but should not touch the skin, in order to prevent burns. A vapour barrier should be added in wet or windy conditions or if wet clothing cannot be removed. Blankets or dry clothing should be placed inside the vapour barrier. A patient with wet clothing who is able to walk should be changed into dry clothing. Walking increases heat production but increases afterdrop. Hypothernic patients found horizontal should not be allowed to walk until after they have been consumed calories and been allowed to shiver for at about 30 min [78,79,80]. 

Rewarming out of hospital can be difficult. Attempts to rewarm should not delay transport. Hypothermic patients are at risk of cardiac arrest (CA) [81]. They should receive adequate oxygenation and be placed on a cardiac monitor [3,38,76]. Defibrillation pads should be applied instead of monitor leads, as they decrease electrocardiographic artifact from shivering allowing rapid detection and treatment of shockable rhythms. While peripheral venous access is desirable, it may be difficult to obtain because of hypothermia-induced peripheral vasoconstriction. If intravenous (IV) access cannot be established, the intraosseous (IO) approach should be used [76]. When intravenous or intraosseous fluids are required, they should be warmed to 38–42 °C and should be given in boluses guided by vital signs [3,77]. Use of heated fluids helps to limit secondary cooling and may protect lines from freezing but has little direct effect on rewarming [77,82].

In moderate or severe hypothermia, bradycardia and low blood pressure do not require specific treatment other than rewarming as they are responses to the global decrease in cellular metabolism [83]. Supraventricular arrhythmias, including atrial fibrillation, usually resolve spontaneously with rewarming [3]. Endotracheal intubation is difficult in cold conditions. IV lines can freeze. Most drugs are ineffective in hypothermia., Endotracheal tubes become less pliable. Endotracheal intubation should usually be deferred until the patient is in a warm environment. Once in a warm environment, rapid-sequence endotracheal intubation should be used. The risk of causing a malignant arrhythmia is minimal compared to the advantages of optimal oxygenation and airway protection [3,39,76]. Rigidity, and trismus in severely hypothermic patients can make intubation difficult [76]. End-tidal carbon dioxide (ETCO_2_) is not reliable in severe hypothermia [84]. Patients should be ventilated using standard weight-based settings without relying on ETCO_2_. Hyperventilation is less detrimental than hypoventilation. Patients should be transported gently and horizontally to avoid triggering hypothermic CA (rescue collapse). 


Hypothermia without vital signs (HT-IV)


Diagnosis of cardiac arrest

It can be difficult to diagnose CA in an unconscious patient in a cold setting. The vital signs may be minimal and extremely difficult to detect. Rescuers should attempt to find vital signs for 60 s [34,36,43]. The use of electrocardiography, ETCO_2_ (any detectable expiratory CO_2_ correlates with the presence of vital signs) or point-of-care ultrasound (POCUS) may help detect organised cardiac activity and significant cardiac output.

Cardiopulmonary resuscitation (CPR)

Chest compressions and ventilation of a hypothermic patient in CA should be performed as for a normothermic patient in CA [34]. If ventricular fibrillation persists after three shocks and the core temperature is <30 °C, further attempts to defibrillate should be delayed until core temperature is >30 °C [34]. Epinephrine and amiodarone should not be given if the core temperature is <30 °C. The administration interval for epinephrine should be doubled from every 3–5 min to every 6–10 min if core temperature is <30 °C. Standard protocols should be resumed once normothermia (≥35 °C) is achieved [34].

Pre-hospital triage

In a hypothermic patient with CA, CPR should not be started or should be terminated if there is a valid “do not resuscitate” order, clear signs of irreversible death, including fixed dependent lividity (livor mortis), danger to or exhaustion of the rescuers, or avalanche burial longer than 60 min with asystole and a completely obstructed airway [3,34,39,76,85]. In a hypothermic patient, rigidity (apparent rigor mortis) is not a reliable sign of death [36]. The following conditions are not contraindications to rewarming of a hypothermic patient in CA: asystole, unwitnessed CA, low core temperature, long no-flow or low-flow time, fixed, dilated pupils, hypocapnia (ETCO_2_ <10 mmHg), old age, or trauma (even major trauma) [38,86,87,88].

Patient referral and transport

The criteria for the direct transfer of a hypothermic patient to an ECLS centre are cardiac arrest, core temperature < 30 °C, systolic blood pressure < 90 mmHg, or ventricular dysrhythmia [34]. Depending on the setting and the distance from an ECLS centre, a stop at a non-ECLS hospital may be used to perform hospital triage, such as the HOPE survival probabilities estimation score www.hypothermiascore.org) [87]. Mechanical CPR should be used for prolonged or technically difficult rescue and transport [34,89]. If transport with continuous CPR is not possible, an alternative is to perform intermittent CPR (Figure 3) [90]. 

When continuous chest compressions are not possible in a patient with a core temperature < 28 °C or if the core temperature is unknown, but the circumstances are unambiguous for hypothermia-induced cardiac arrest, alternate at least 5 min of CPR with ≤5 min without CPR. In patients with a temperature < 20 °C, alternate at least 5 min of CPR with ≤10 min without CPR [90].

Once a patient is in hypothermic cardiac arrest, avoiding rapid movement is no longer necessary. Although it is reasonable to transport a patient with ongoing CPR in a horizontal position, there are insufficient data available to make definite recommendations [91]. Rapid transport even in a non-horizontal position is preferable to slower transport in a horizontal position.

### 5.2. In-Hospital Treatment

In-hospital treatment options depend on circulatory status, the stage of hypothermia and available resources (Figure 4). 

Patients with stable circulation should be rewarmed with passive and active external rewarming techniques (Table 5). In patients at risk of cardiac arrest or with unstable circulation, ECLS should be kept on standby and should be initiated rapidly if needed. Patients in hypothermic CA or with hemodynamic instability require cardiocirculatory support in addition to active internal rewarming. This is best achieved with ECLS (Table 5). 

In patients with spontaneous circulation, and a core temperature of 33–36 °C the goal of rewarming should be normothermia with a core temperature about 37 °C. In patients in hypothermic cardiac arrest, the first priority is return of spontaneous circulation (ROSC), because physiological organ perfusion improves oxygen delivery and reduces no-flow and low-flow organ damage during CA compared to CPR. Once ROSC is established, the goal should be targeted temperature management according to local protocols.

Rewarming is passive if the patient increases the core temperature without exogenous heat, and active if exogenous heat is delivered. Active rewarming can be external oe internal, depending on the method of heat transfer (Table 5). Most patients require passive and active external rewarming. Patients in cardiac arrest, and those who are unable to rewarm with active external rewarming require active internal rewarming (Table 5). 

In mild hypothermia (HT I), patients should be rewarmed using passive and active external rewarming and with minimally invasive internal rewarming using warm fluids orally (Figure 4). Shivering and active movement will further speed rewarming. Shivering should be allowed if there are no contraindications, such as pain from injuries or risk of myocardial injury caused by increased oxygen consumption in a patient with high cardiovascular risk.

In patients with moderate or severe (HT II or HT III) ), active rewarming is necessary. Usually, active external rewarming is effective. If a patient fails to rewarm adequately, active internal rewarming should be used. Signs of failure to rewarm include core temperature remaining the same or decreasing, increasing lactate levels, decreasing level of consciousness, decreasing blood pressure, or the occurrence of ventricular dysrhythmias. Active internal rewarming methods in moderate to severe hypothermia include intravascular rewarming catheters and continuous renal replacement therapy (CRRT, Table 5). Complications, such as bleeding, deep venous thrombosis, and infection are more common than with active external rewarming. 

Patients without vital signs (HT IV) are in cardiac arrest and require cardiocirculatory support. The preferred method is extracorporeal life support (ECLS) rewarming [88]. The ideal rewarming rate is not known. The target rewarming rate should be ≤5 °C/h [94]. A slower rate of rewarming (approximately 2 °C/h) may be associated with improved survival with good neurologic outcome [131]. Extracorporeal membrane oxygenation (ECMO) is preferred to cardiopulmonary bypass (CPB), because ECMO can be maintained post-ROSC and may be used to maintain life support while adult respiratory distress syndrome (ARDS), a common sequela of hypothermia, resolves. [132]. Non-ECLS rewarming should only be considered when ECLS rewarming is not available and cannot be provided within 6 h. The ideal non-ECLS rewarming bundle has not been determined. Rewarming should use locally available techniques that are familiar to the medical staff, such as heating blankets, external forced air rewarming, and peritoneal lavage [3]. Although IV fluids are not effective in rewarming, IV fluids should be warmed to 40 °C to avoid cooling. CPR should be maintained continuously until ROSC is achieved. Mechanical chest compression can be helpful in long-duration CPR. The team should be trained to avoid prolonged hands-off times and to prevent injury to the patient [34].

In moderate and severe hypothermia, circulating blood volume is decreased because of a combination of extravascular plasma shift and inadequate fluid intake. In patients with cold exposure lasting several hours, cold-induced diuresis may further deplete the intravascular volume. Patients immersed for hours may have hydrostatic pressure induced diuresis and may be severely fluid depleted [76]. At the same time, circulatory centralisation may decrease the vascular space. Volume replacement should be given carefully to avoid overload. Ideally, volume replacement should be given with central venous pressure monitoring.

During rewarming, hypothermia-induced vasoconstriction that previously limited the vascular space is abolished. Volume should be replaced to avoid severe volume depletion with resultant shock. Blood gas interpretation should be performed according to the alpha-stat approach, evaluating blood gas samples at 37 °C. This leads to improved cerebral perfusion and better neurologic outcome compared to using temperature corrected arterial blood gas values [133,134]. Some studies suggest that in severely hypothermic patients the PaCO_2_–ETCO_2_ gradient is increased. This may help to confirm a diagnosis of accidental hypothermia in unclear circumstances [84]. Several tools have been developed to help predict outcome [132]. The Hypothermia Outcome Prediction after Extracorporeal Life Support (HOPE) score—is currently the best available prediction tool and should be used to evaluate whether a hypothermic patient in cardiac arrest is likely to benefit from ECLS rewarming [87,135]. Frostbite injuries that are still frozen should be thawed as soon as circulation and core temperature have been stabilised [136,137,138]. 

#### Extracorporeal cardiopulmonary resuscitation

Extracorporeal cardiopulmonary resuscitation (ECPR) [139] followed by ECLS rewarming may be associated with higher survival and more favourable neurological outcomes than conventional CPR alone in patients with accidental hypothermia without vital signs [121,126]. Rapid technological progress including the miniaturisation and improved efficacy, safety, and transportability of ECLS devices are expanding the possibilities of ECPR and ECLS rewarming [122,140]. 

Good neurological outcomes are possible, even with prolonged no-flow or low-flow times, but only if hypothermia developed before cardiac arrest [38,88,125,126,131,141]. Because of the low rates of survival and the high likelihood of major morbidity in unselected patients, ECPR should not be used if core temperature is >30 °C, there is major trauma, or there are comorbidities that are not compatible with an acceptable quality of life if the patient recovers [87,126,142,143]. ECPR, provided in a centre with expertise and streamlined protocols, can improve survival and neurological outcomes [144]. 

Veno-arterial (VA) ECMO is the preferred method of ECPR because anticoagulation requirements are minimal and because VA ECMO may be able to provide circulatory and respiratory support beyond ROSC [88,139,145]. The chances of survival are higher after rewarming with ECMO than with CPB. The chances of survival are also higher in patients with witnessed hypothermic cardiac arrests than in those with unwitnessed hypothermic CAs. Avalanche victims have the lowest probability of surviving. Male sex, high initial body temperature, low pH, and high serum potassium were associated with decreased chances of survival [121,146]. 

Practical approaches to patient management with ECPR and ECLS rewarming have been described [88,131,147]. Additional measures, such as vigorous resuscitation with isotonic fluids and the use of vasopressors are usually needed to maintain mean arterial pressure between 50 and 70 mmHg. Severe acid-base and coagulation disturbances and other problems may require additional treatment [88,145]. The core temperature should be carefully monitored (Table 4) [88,148]. 

Patients in hypothermic cardiac arrest should receive continuous ECPR until ROSC is achieved. Further support with ECLS after ECPR may be warranted. However, the decision to continue care without signs of immediate significant improvement is difficult [149]. Predictors of survival have been described in retrospective studies [87,131,150] Lower initial serum lactate, higher pH, and younger age were associated with better neurological outcomes in survivors of hypothermic OHCA treated with ECPR [88,131,150,151]. There is currently no simple, reliable predictor of ECPR efficacy or outcome. Targeted temperature management (TTM) should be performed according to local protocols. Post-resuscitation hyperthermia should be avoided [145]. The timing of ECPR and the optimum target temperature following ECPR, are unknown [152,153].

Termination of ECPR should be considered if there is no ROSC at the target core temperature. The decision to stop treatment may also be based on additional clinical factors, such as the onset of uncontrollable hemorrhage, further information about the cause of CA, or signs of severe anoxic brain injury [3,154]. Brain death is a common cause of death following ECPR. Organ donation should be considered [139,154].

## 6. Outlook

Compared to other conditions that may result in cardiac arrest, accidental hypothermia is rare. Even large centres rarely treat more than 20 patients annually. The largest studies to date have included only a few hundred patients [40]. Pooling of data is important to hasten the development of methods for diagnosis, treatment, and outcome prediction. The International Hypothermia Registry has already collected data from more than 200 patients [155]. Further contributions are necessary and are welcome. 

A patient with mild hypothermia found in a horizontal position should rest and should be allowed to shiver before being allowed to stand up or to walk [76], but some guidelines suggest that, if the patient is able to move safely, immediate physical exertion hastens rewarming and facilitates terrestrial evacuation in difficult terrain [3,79]. The in-hospital HOPE score has substantially improved outcome prediction for hypothermic patients in cardiac arrest who are treated with ECLS rewarming. A prehospital HOPE score might facilitate treatment and transport decisions in the field. When ECLS rewarming is not available, non-ECLS measures should be used. No data currently exist regarding the efficacy and safety of various non-ECLS measures. High-quality studies on non-ECLS rewarming in hypothermic patients in CA are urgently required.

## 7. Conclusions

Accidental hypothermia is an unintentional drop of core temperature below 35 °C. Annually, thousands die of primary hypothermia and an unknown number die of secondary hypothermia worldwide. Hypothermia can be expected in emergency patients in the prehospital phase. Injured and intoxicated patients cool quickly even in subtropical regions. Preventive measures are important to avoid hypothermia or cooling in ill or injured patients. Diagnosis and assessment of the risk of cardiac arrest are based on clinical signs and core temperature measurement when available. Hypothermic patients with risk factors for imminent cardiac arrest (temperature < 30 °C in young and healthy patients and <32 °C in elderly persons, or patients with multiple comorbidities), ventricular dysrhythmias, or systolic blood pressure < 90 mmHg) and hypothermic patients who are already in cardiac arrest, should be transferred directly to an extracorporeal life support (ECLS) centre. If a hypothermic patient arrests, continuous cardiopulmonary resuscitation (CPR) should be performed. In hypothermic patients, the chances of survival and good neurological outcome are higher than for normothermic patients for witnessed, unwitnessed, and asystolic cardiac arrest. Mechanical CPR devices should be used for prolonged rescue, if available. In severely hypothermic patients in cardiac arrest, if continuous or mechanical CPR is not possible, intermittent CPR should be used. Rewarming can be accomplished by passive and active techniques. Most often, passive and active external techniques are used. Only in patients with refractory hypothermia or cardiac arrest are internal rewarming techniques required. ECLS rewarming should be performed with extracorporeal membrane oxygenation (ECMO). A post-resuscitation care bundle should complement treatment.

## Figures and Tables

**Figure 1 ijerph-19-00501-f001:**
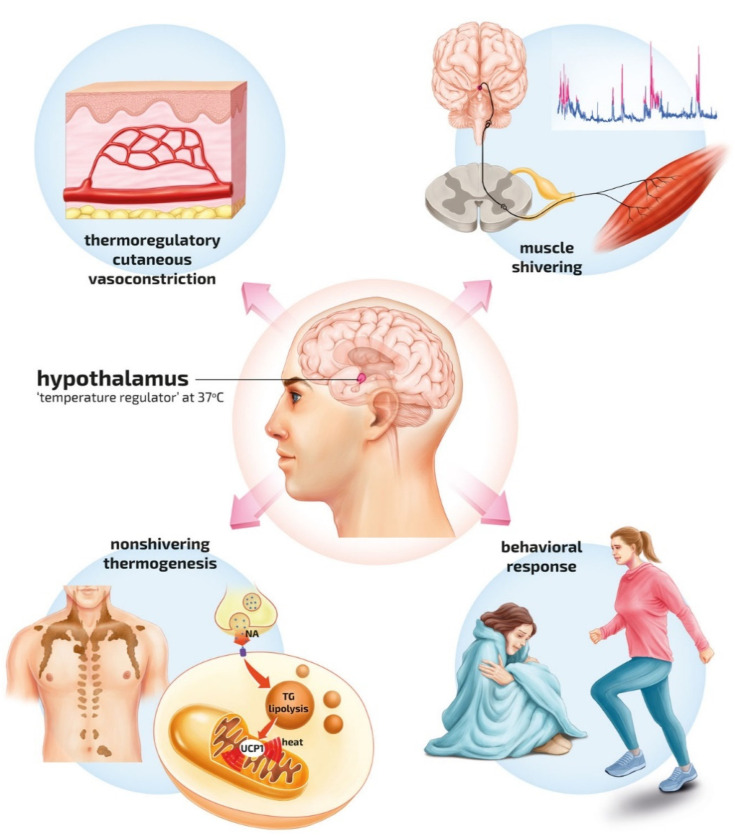
Physiologic pathways of central and peripheral thermoregulation against environmental cold. Increased sympathetic tone causes cutaneous vasoconstriction reducing skin blood flow, decreasing heat loss. Vasoconstriction increases insulation of tissue, reducing conductive heat transfer, and minimising exposure of warm blood to the cold environment (top left). Shivering thermogenesis in skeletal muscles provides endogenous heat production [19]. The contribution of shivering to heat production depends on the strength of the cold stimulus. The stronger the stimulus, the more intense the heat production. The intensity of shivering also depends on the dominant pattern of shivering, continuous versus burst shivering, and the availability of energy substrates, mainly glucose. Involuntary shivering can counteract cooling by increasing the endogenous basal heat production up to 500% of baseline (top right) [20]. Non-shivering thermogenesis occurs in brown adipose tissue [21]. The source of non-shivering thermogenesis is primarily the uncoupling of oxidative phosphorylation This is accomplished by a mitochondrial proton leak via uncoupling protein 1 (UCP1) in the unacclimatised human and an increase in brown adipose tissue thermogenesis following cold acclimatisation. UCP1 creates a proton leak across the inner mitochondrial membrane, diverting protons away from ATP synthesis and resulting in heat production (lower left). Behavioural responses are somatic motor acts primarily directed toward minimising heat loss or generating endogenous heat. Exercise-induced thermogenesis provides the greatest heat gain, reaching values up to 15–20 times above the resting metabolic rate. Exercise in cold conditions may not be advisable, as it carries a risk of overexertion leading to further cooling and circulatory collapse (lower right).

**Figure 2 ijerph-19-00501-f002:**
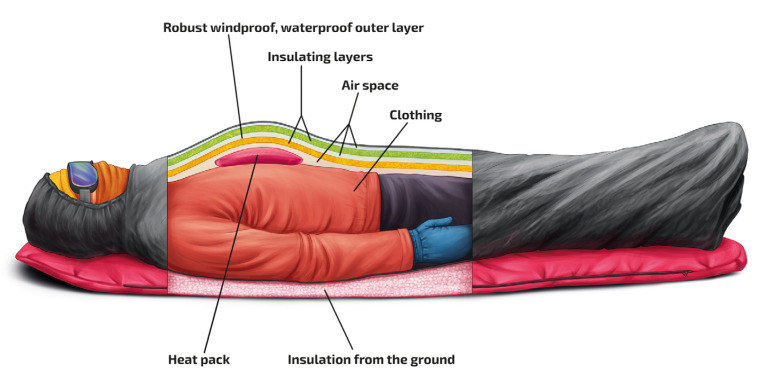
Improvised insulation without a commercial hypothermia bag should consist of an outer robust windproof, and waterproof vapour-barrier cover. Inside the cover, blankets can be used for insulation. Chemical or electric heat packs can be placed on the trunk but should not be applied directly to the skin. Mittens (or gloves if mittens are not available) should be placed on the hands. The head, including the face, and the neck should be protected against the cold.

**Figure 3 ijerph-19-00501-f003:**
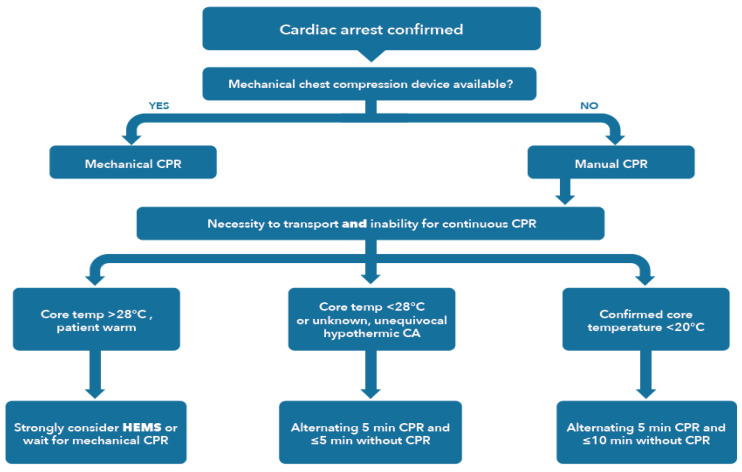
Intermittent CPR algorithm for severely hypothermic patients (<28 °C) in cardiac arrest when continuous chest compressions are not possible. CRITICAL CORRECTIONS: 1. The statement ‘patient warm,’ does not mean that the patient is warm. It means that the patient is not sufficiently hypothermic for intermittent CPR. 2. ‘Alternating 5 min CPR’ and ≤5 min or ≤10 min without CPR should read: ‘Alternating at least 5 min CPR and ≤5 min or ≤10 min without CPR. Reprinted with permission from [34]. Copyright 2021 Elsevier and European Resuscitation Council.

**Figure 4 ijerph-19-00501-f004:**
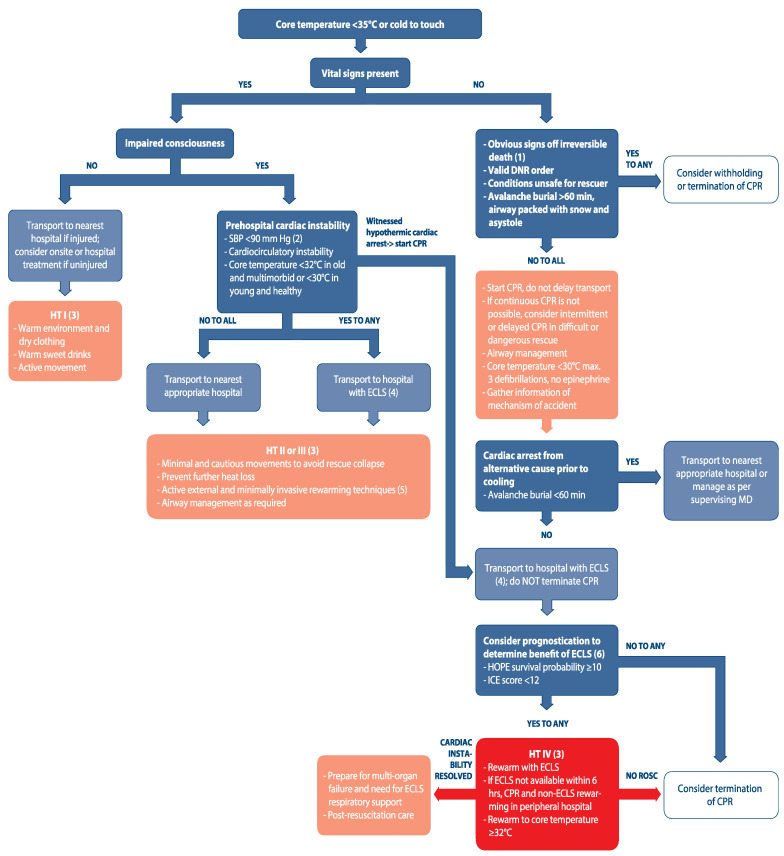
Accidental hypothermia treatment algorithm. Reprinted with permission from [34]. Copyright 2021 Elsevier and European Resuscitation Council.

**Table 1 ijerph-19-00501-t001:** Conditions associated with secondary hypothermia, adapted from [3]. Subtitles are marked in italics. Reprinted with permission.

Impaired Thermoregulation	Decreased Heat Production	Increased Heat Loss
*Central nervous system failure*	*Endocrine failure*	*Dermatologic lllness*
Anorexia nervosa	Alcoholic or diabetic ketoacidosis	Burns
Stroke	Hypoadrenalism	Induced vasodilation
Traumatic brain injury	Hypopituitarism	Medications and toxins
Hypothalamic dysfunction	Lactic acidosis	
Metabolic failure		*Iatrogenic*
Neoplasm	*Insufficient fuel*	Emergency childbirth (possibly without prevention of hypothermia)
Parkinson’s disease	Extreme physical exertion	Cold infusions
Pharmacologic effects (anaesthetic drugs)	Hypoglycaemia	Heat-stroke treatment
Stroke, haemorrhagic or ischaemic	Malnutrition	
Toxins		*Other associated clinical states*
	*Neuromuscular compromise*	Carcinomatosis
*Peripheral failure*	Extremes of age	Cardiopulmonary disesae
Acute spinal cord transection	Impaired shivering	Major infections
Peipheral neuropathy	Inactivity	Multiple trauma
		Shock

**Table 2 ijerph-19-00501-t002:** Classical staging of accidental hypothermia based on clinical signs [34]. Reprinted with permission. Copyright 2021 European Resuscitation Council.

Stage	Clinical Findings	Estimated Core Temperature ( °C)
Hypothermia I (mild)	Conscious, shivering *	35–32 °C
Hypothermia II (moderate)	Impaired consciousness *; may or may not be shivering	<32–28 °C
Hypothermia III (severe)	Unconscious *; vital signs present	<28 °C
Hypothermia IV (severe)	Apparent death; vital signs absent	Classically < 24 °C **

* Shivering or consciousness may be impaired by comorbid conditions such as trauma, central nervous system conditions, toxins or drugs, such as sedative-hypnotic drugs or opioids, independent of core temperature. ** Cardiac arrest can occur at earlier or later stages of hypothermia. Some patients may have vital signs with core temperatures < 24 °C.

**Table 3 ijerph-19-00501-t003:** Principles of pre-hospital management of hypothermia, according to the revised Swiss system [41]. AVPU: alert, verbal, responsive, unconscious [44]; ECLS: extracorporeal life support. CPR: cardiopulmonary resuscitation. ‘+’ means recommended; ‘−‘ means not recommended. Reprinted with permission from [41]. Copyright 2021 Elsevier and European Resuscitation Council [41].

	Stage 1	Stage 2	Stage 3	Stage 4
Clinical findings ^1^	“Alert” from AVPU	“Verbal” from AVPU	“Painful’’ or “Unconscious” from AVPU Vital signs present	“Unconscious” from AVPU AND No detectable vital signs ^2^
Risk of cardiac arrest ^3^	**Low**	**Moderate**	**High**	** Hypothermic cardiac arrest **
Oxygen according to ususal clinical practice, (goal: SpO_2_ > 94%) ^4^	+	+	+	+
Carbohydrates	Warm sweet tea, sweet bars	Glucose IV/IO. ^5^	Glucose IV/IO. ^5^	−
Active movement	+	− ^6^	−	−
Passive rewarming	+	+	+	+
Active rewarming	(+)	+	+	+
Cautious mobilisation/horizontal transport if possible	−	+	+	−
Defibrillation pads	-	+	+	+
Intubation	-	-	Consider	+
CPR	-	-	-	+
Defibrillation	-	-	-	+ ^7^
Drugs (CPR)	-	-	-	+ ^8^
Hospital with ECLS ^9^	-	-	+	+

^1^ In the revised Swiss system, “Alert” corresponds to a GCS score of 15. “Verbal” corresponds to GCS scores of 9–14, including confused patients. “Painful” and “Unconscious” correspond to GCS scores < 9. While shivering is not used as a stage-defining sign in the revised Swiss system, its presence usually means that the temperature is >30 °C, a temperature at which hypothermic CA is unlikely to occur in healthy patients. ^2^ No respiration, no palpable carotid or femoral pulse, no measurable blood pressure. Check for signs of life (pulse and, especially, respiration) for up to 1 min. ^3^ The transition of colours between stages represents the overlap of patients within groups. The estimated risk of cardiac arrest is based on accidental hypothermia being the only cause of the clinical findings. If other conditions impair consciousness, such as asphyxia, intoxication, high altitude cerebral oedema or trauma, the revised Swiss system may falsely predict a higher risk of cardiac arrest due to hypothermia. Caution should be taken if a patient remains “alert” or “verbal” showing signs of haemodynamic or respiratory instability such as bradycardia, bradypnoea, or hypotension because this may suggest transition to a stage with a higher risk of cardiac arrest. ^4^ Might be difficult to measure because of peripheral vasoconstriction. ^5^ Glucose should be given for hypoglycaemia. If point-of-care glucose testing is not available, glucose can be given empirically to a hypothermic patient with altered mental status. ^6^ Active movement allowed if distinct shivering is present and the patient is already standing or ambulating.^7^ If ventricular fibrillation persists after three shocks and the temperature is <30 °C, delay further attempts of defibrillation until temperature > 30 °C. ^8^ Withhold epinephrine (adrenaline) and amiodarone if temperature < 30 °C; increase interval of administration to 6–10 min for epinephrine if temperature 30–35 °C. ^9^ In addition to patients in cardiac arrest, patients with core temperatures < 30 °C, systolic blood pressure < 90 mmHg, or ventricular dysrhythmias, should be transferred directly to an extracorporeal life support (ECLS) centre; (+) means can be considered.

**Table 4 ijerph-19-00501-t004:** Measurement of core temperature in hypothermia, from least to most invasive.

Type of Measurement	Characteristics	Limitations	Suitability for Core Temperature Measurement in Hypothermia	Feasible in Hospital (IH) or Out of Hospital (OH)	References
Touching the skin of torso torso	No equipment needed High negative predictive value	Rough estimate, only Not validated for use in cold environments	(+)	(OH)/IH	[57]
Temporal artery (infrared)	Rapid, non-invasive, convenient, low cost, hygienic measurement	Serious time lag during cooling and rewarming Strongly affected by ambient temperature, positioning and vasomotor activity Low accuracy	-	None	[50,56,58,59,60,61,62]
(Forehead) skin (infrared radiation, electronic thermistor, liquid crystal strip)	Rapid, non-invasive, convenient, low-cost, hygienic measurement	Can be several degrees lower than core temperature Highly influenced by ambient temperature	-	None	[47,50,54,63,64]
Temporal microwave thermometer	Rapid, non-invasive, easy to use Correlates well with brain temperature, even in severe hypothermia	Still experimental, but promising	(+)	IH	[46]
Zero-heat-flux thermometer on the forehead. Deep tissue temperature is measured at the skin by an insulated temperature probe)	Rapid, non-invasive, convenient measurement Good agreement with core temperature in normothermia and mild hypothermia	Equilibration takes several minutes Poor agreement with decreasing core temperature (not reliable below 34 °C) Has not been tested in cold environments	-	IH	[42,47,64,65,66]
Axillary (electronic device or glass thermometer ^+^)	Rapid, non-invasive, hygienic, convenient measurement	Strongly affected by ambient temperature and positioning Reading is lower than in other locations Significant time lag during cooling or rewarming Low accuracy	-	None	[50,54,56,60]
Tympanic (infrared radiation)	Rapid, non-invasive, hygienic, convenient measurement	Inaccurate in hypothermic patients Inaccurate in cold and hot environment, if incorrectly positioned, with otitis media (hyperaemia and inflammation), if the tympanic membrane is blocked by ear wax, or with water or snow in the external auditory canal Low accuracy	-	(IH)	[42,50,52,56,58,62,67,68,69]
Epitympanic (electronic thermistor)	Good correlation with arterial blood temperature, even in rapid cooling and rewarming Reliable for non-intubated patients in out of hospital use Strong correlation with brain temperature in several studies	Requires open extrenal auditory canal, good insulation, and fixation initially requires a few minutes to stabilise Not widely available Influenced by head and neck temperature Inaccurate during cardiac arrest Not as accurate as bladder or rectal core temperature measurement during steady state	+	OH/IH	[42,48,49,51,53,55,70,71]
Oral (electronic thermistor or glass thermometer ^+^)	Rapid, non-invasive, hygienic, convenient measurement	Influenced by positioning, breathing with open mouth Not accurate in hot and cold environment Low accuracy Influenced by head and cervical temperature	-	IH	[70]
Nasopharyngeal (electronic thermistor)	Rapid, minimal invasive Estimates brain temperature if placed approximately 10–14 cm deep	Only in sedated or anaesthetised patients False readings during cooling and rewarming because adherence to adjacent tissue not assured	+	OH/IH	[72]
Gastrointestinal temperature (telemetry temperature sensor)	Higher validity compared to rectal measurement	Slower response to changes than oesophageal measurement Experimental Unpredictable location. Impractical. Must be ingested 4 to 8 h before use	-	None	[50,72]
Oesophageal (electronic thermistor)	Lower third of oesophagus (approx. 40 cm insertion depth from the incisors) Good correlation with arterial blood temperature, especially in steady state Standard for out-of-hospital intubated patients	Inaccurate values during open chest surgery with cardiac cooling Insertion of probe may provoke vomiting and aspiration, nasal bleeding, cardiac arrhythmias, and cardiac arrest. Can be misplaced in the trachea. Relatively contraindicated in patients with unsecured airways.	+	OH/IH	[42,48,54,56,63,73,74]
Bladder (electronic thermistor)	Close correlation with arterial blood temperature in steady state Reliable core temperature for in-hospital use, widespread use Cand be used in combination with monitoring of urine output.	Lags during cooling or rewarming, although less than with rectal measurement) Influenced by urine output (Cold diuresis increases urine output.) Reasonable if urinary catheter required. May be embarrassing for the patient. Care needed to place hygienically.	+	IH	[42,47,48,51,56,62,71]
Rectal (electronic thermistor or glass thermometer ^+^)	Close correlation with arterial blood temperature in steady state	Inaccurate if placed into stool. Probe should be inserted 15 cm past rectum Significant lag time during cooling and rewarming. May be embarrassing for the patient. Non-hygienic Possible perforation of the rectum	+	IH	[42,48,51,54,56,63,71]
Pulmonary artery catheter (electronic thermistor)	Directly measures the temperature of blood leaving the heart. Defines core temperature.	Not available out of hospital or in many hospitals Very invasive, with potential for severe complications.	+	IH	[42,47,48,54,56]
Brain temperature	Measures the temperature of the brain	Good correlation with core temperature Only possible in experimental settings or during neurosurgery In general, it is difficult to track brain temperature with other monitoring sites.	+	IH	[48,51,54,63]

^+^ Thermometers filled with mercury or other liquids can no longer be purchased. Heat is infrared radiation from 300 GHz to 1 THz. Microwave radiation has a frequency of 1 GHz to 300 GHz; (+) means can be considered, ‘+’ reasonable, ‘-‘ not reasonable.

**Table 5 ijerph-19-00501-t005:** Rewarming techniques in patients with accidental hypothermia [32].

Rewarming Technique	Rewarming Rate	Notes & Controversies	Rewarming Complications
Passive Rewarming			
Passive rewarming [3,92]	0.5–4 °C /h (dependends on patient’s thermoregulatory function and metabolic reserves)	Protects from further heat loss and allows patient to self-rewarm.	Negligible in isolated mild hypothermia. For colder patients and those with secondary hypothermia or comorbidities, passive rewarming alone is not adequate.
Passive rewarming with active movement [93]	1–5 °C /h	Exercise immediately after rescue increases afterdrop	Increased afterdrop could cause rescue collapse.
Active External Rewarming			
Active rewarming including forced-air surface rewarming [94,95], heating pads, e.g. Arctic Sun^®^ [96,97,98],warmed IV fluids (40 °C).	0.5–4 °C /h	Protects from further heat loss, delivers external heat. Warmed IV fluids are not effective if used as the sole method of rewarming.	Similar to passive rewarming
Active Internal Rewarming			
Bladder lavage [31,92,99]	Variable. Adds < 0.5 °C /h	Not recommended Rewarming is intermittent and slow because of small surface area. Poor control of infusate temperature	Negligible unless difficult catheterisation
Gastric lavage [31]	May add ~0.5–1 °C /h	Not recommended. Unacceptably high risk to benefit ratio	Potential for aspiration, fluid and electrolyte shifts
Intravascular catheter rewarming, e.g., CoolGuard^®^ [36,100,101,102] Quattro^®^ [103] Cool Line^®^ [104] Innercool^®^ [105]	Device specific (adds ~0.5–2.5 °C /h)	Uncertain indications for use. Potential beneficial for colder patients, especially those with comorbidites, with stable circulation	Potential for haemorrhage or thrombosis, potentially worsening arterial hypotension in unstable patients
Thoracic [106,107] or peritoneal lavage [108,109]	Variable, depending on tempearture and flow rate of paricardial irrigation.	May be useful in unstable patients when ECLS rewarming is not available. Very invasive.	Potential for haemorrhage, lung or bowel trauma, fluid and electrolyte shifts. Thoracic lavage may interfere with CPR
CRRT (including CVVHF, CVVHD, CVVHDF) [92,110,111,112,113]	Adds ~1.5–3 °C /h	Not recommended unless ECLS rewarming not available. Require adequate blood pressure. Heparinisation, citrate anticoagulation, or prostacyclin required	Problems rare. Local vascular complications. Air embolism. Arterial hypotension
Haemodialysis [92,114,115,116,117,118]	Adds ~2–3 °C /h	Patient must be able to increase cardiac output to perfuse the external circuit. Heparinisation required	Potential for arterial hypotension, haemorrhage, thrombosis, haemolysis, etc.
Veno-venous rewarming (usually with ECMO) [99,119]	~4–10 °C /h	Provides no circulatory or ventilatory support in case of cardiac arrest. Patient must be able to increase cardiac output to perfuse the external circuit	Potential for arterial hypotension, haemorrhage, thrombosis, haemolysis, etc.
Extra-corporeal life support (ECLS; VA-ECMO, CPB including minimally invasive extracorporeal circulation (MiECC)) [83,120,121,122,123,124,125,126,127,128,129,130]	~4–10 °C /h	Preferred rewarming method for patients in cardiac arrest. ECMO preferred over CPB. ECMO can use femoral route avoiding need for sternotomy. Can be used to treat post-rewarming pulmonary complications, such as ARDS.	Potential for haemorrhage and arterial hypotension, thrombosis, haemolysis, etc., as with all intravascular devices

CPB: cardiopulmonary bypass, CPR: cardiopulmonary resuscitation, ECLS: extracorporeal life support, ECMO: extracorporeal membrane oxygenation, CRRT: continuous renal replacement therapy, CVVHF: continuous veno-venous haemofiltration, CVVHD: continuous veno-venous haemodialysis, CVVHDF: continuous veno-venous haemodialysis with filtration. Adapted with permission from [3]. Copyright 2016 ICAR MedCom.

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
