# Peer review of "Accidental Hypothermia: 2021 Update"

_ijerph, 2022, doi:10.3390/ijerph19010501_

Round 1

Reviewer 1 Report

Reviewer comments:

I am pleased to have the opportunity to review your paper. I believe that an updated review on hypothermia would be very useful for many readers. I would like to ask for your comments on a few points.

Major points:

2.Epidemiology

Line 76-77, Is it possible that the reason for the range in the frequency of hypothermia is related to the lack of epidemiological studies on hypothermia itself and the lack of nationwide studies?

3.Pathophysiology

Line 99-100, “Nonshivering thermogenesis occurs in brown adipose tissue and in skeletal muscle.”

I understand the heat production that occurs in skeletal muscle is shivering. Is this sentence in need of revision?

Line 98-99, “The maximal rate of heat produced involuntarily is approximately five times the resting metabolic rate.”

Line 151-153, “Shivering counteracts cooling by increasing endogenous heat production by as much as 500% of basal metabolism.”

The two sentences are duplicates of the same content and are expressed with different phrases, so the authors should use consistent expression and revise the description.

Line 158, “In young, healthy adults, hypothermia-induced cardiac arrest may occur below 30°C.”

Line 176-177, “At core temperatures bove 30 °C no cardiac arrest was solely due to hypothermia.”

Is there any overlapping content?

5.Treatment

Line 96-97, “The use of electrocardiographic or ETCO2 monitoring or a point-of-care ultrasound (echocardiography) may help to detect minimal, but perfusing, cardiac activity.”

Doesn't that contradict what you wrote about ETCO2 monitoring being not reliable (line 96-97)?

Line 206-208, “In severely hypothermic patients the PaCO2 - ETCO2 gradient may be increased. In unclear circumstances an increased PaCO2 - ETCO2 gradient may suggest the diagnosis of accidental hypothermia.”

Is this perhaps not appropriate for description in the treatment section? Isn't the description rather a diagnosis?

Minor points:

There were some minor spelling and grammatical errors, so please check and correct them again.

1.Introduction

Line 47, ccidental hypothermia → accidental hypothermia

3.Pathophysiology

Line 128, ultiple organ system → multiple organ system

Line 134, ~alone In such patients → ~alone. In such patients

Line 172 teh → the

4.Diagnosis of Hypothermia

Line 198 the .brain → the brain

Line 203 ~quirements All devices → ~quirements. All devices

Line 203 ~variations. [45]. → ~variations [45].

Line 10 such as. microwave → such as microwave

Line 13 ~assessment at can → ~assessment can

5.Treatment

Line 200 ~vascular space.  → ~vascular space,

Line 206 Thi → This?

Line 232 ROSC) → ROSC

7.Conclusions

Line 291 ~is not possible. → ~is not possible, 

Line 292 active.. → active.

Line 294 if available.. → if available.

Table5.

Rewarming Complications

Potential for haemorrhage or thrombosis Could worsen hypotension in unstable patients.

As with all intravascular devices).

Author Response

Please see the attachment, thank you

Reviewer 2 Report

This 2021 updated review on Accidental Hypothermia, by Paal et al, is an extensive overview of the topic. I offer a few minor suggestions and clarifications for improvement:

Introduction: Can you please clarify the use of “accidental hypothermia” versus just “hypothermia”? What are other types of hypothermia and why the difference (i.e., there is probably some distinction with medically induced hypothermia to treat specific medical conditions)

Table 1: what is “emergency childbirth” compared with non-emergency childbirth? Please clarify

Page 3, line 59 – remove extra period

Page 5, line 117 – “multipe” should be “multiple”

Page 5, lines 121 and 130 – please spell ot and define “ISS” score

Page 5, line 128 – “ultiple” should be “multiple”

Page 5, line 139 – frostbite should be singular

Page 6, line 172 – “the” should be “the”

Page 6, line 176 – “bove” should be “above”

Page 7, line 191 – “hyothermia” should be “hypothermia”

Table 3 is highly confusing. What do the plusses and minuses refer to? What is “AVPU”? Carbohydrates is misspelled. Please revise and place on a single page for clarity.

Unknown page, line 198 – remove extra period (.brain?)

Unknown page, line 40 – “Packagining” is not correct

Page 3 of 37, line 113 – “no contraindications” is confusing as written

Figure 3 – what is “HEMS”?

Figure 4 is so small I can’t read it. PLEASE REVISE AND MAKE MORE LEGIBLE

Table 3, line 150 – “updated from [33]” is an incomplete sentence

Page 10 of 37, line 261 and 278 – in the first sentence of “Outlook”, the authors state Accidental hypothermia is “rare”. However, in the first sentence of the Conclusion, the authors state “thousands die” worldwide. These two opening statements appear contradictory as written.

Author Response

Please see the attachment, thank you very much
